# Pre-colonial Amerindian legacies in forest composition of southern Brazil

**Aline Pereira Cruz** [ID]¹*, **Eduardo Luiz Hettwer Giehl** [ID]¹, **Carolina Levis**¹, **Juliana Salles Machado**², **Lucas Bueno**², **Nivaldo Peroni**¹

**1** Department of Zoology and Ecology, Graduate Program in Ecology, Federal University of Santa Catarina, Florianopolis, Santa Catarina, Brazil, **2** Department of History, Federal University of Santa Catarina, Florianopolis, Santa Catarina, Brazil

* cruz.p.aline@gmail.com

**Data Availability Statement:** We have organized the data in order to preserve the third-party data, but to allow the analysis to be reproduced. The following data is in the supporting information files: All steps of the analysis in R, containing the names

## Abstract

Past human societies have left persistent marks on forests worldwide. However, the degree to which pre-colonial Amerindian societies have affected forest structure is still not fully understood, especially in southern Brazil. This study investigated the influence of two distinct Amerindian groups (Southern-Jê and Guarani) over tree composition of forest fragments in the State of Santa Catarina. Vegetation data was obtained from the Santa Catarina Forest and Floristic Inventory (SCFFI): a statewide systematic vegetation sampling project. Archaeological data was collated from literature reviews as well as existing databases for archaeological sites occupied by Guarani and Southern-Jê groups. Using these sites of known Amerindian occupation, and corresponding environmental variables, ecological niche models were developed for each Amerindian group, predicting potential archaeological sites occupied by these groups across southern Brazil. Maps of these potential occupation sites of pre-colonial Amerindian groups were compared with 417 corresponding floristic inventory plots. Redundancy analysis (RDA) was used to identify floristic composition patterns linked to areas with a high probability of Southern-Jê or Guarani presence. Southern-Jê and Guarani pre-colonial occupations overlapped near main rivers; however, Southern-Jê groups generally occupied elevated areas whereas Guarani occupied mostly coastal areas. We observed differences in forest composition associated with the predicted occurrence of these pre-colonial Amerindian groups. Based on these results, we argue there is a relationship between tree species distribution and pre-colonial human occupation by these two Amerindian groups.

## Introduction

To understand species distribution, we need to consider many different aspects, both biotic and abiotic, and their interdependence, as well as the historical processes. Historical ecology research seeks to understand past human legacy on present day species composition, community assemblage, and identify positive interactions (eg. mutualism) and process associated with niche modification [1–3]. A place with the appropriate environmental characteristics for a

of the files we are sharing and / or the download paths. Two georeferenced images, with the results of niche modeling, The Guarani model and the Southern-Jê model. Part of the geographical coordinates of the Southern-Jê archaeological sites. Floristic data are third-party Data. We are sharing the abundance matrix with anonymization of species names and sample units. These data belong to the Floristic Forest Inventory of the State of Santa Catarina (IFFSC), and are published in books, available for download at: <https://sites. google.com/view/iffportal/publica%C3%A7%C3% B5es/livros>. We requested the abundance matrix (raw data) from the IFFSC coordination, through an email with a description of our research objective. The project coordinator is Dr. Alexander Christian Vibrans, from the Regional University of Blumenau - FURB. Their emails are: acv@furb.br; iffsc@furb. br; inventarioflorestal.sc@gmail.com. The portion of the Southern-Jê data that is published in Noelli e Souza, 2017, (doi: 10.1590/1981. 81222017000100004) was provided by the authors and is third party data and, for that reason we are not sharing. However, we are sharing a set of locations from Southern-Jê archaeological sites. All these locations are in a radius of 200 m. from some archaeological site published by the National Historical and Artistic Heritage Institute of Brazil (IPHAN) in: http://portal.iphan.gov.br/cna/pagina/ detalhes/1227 data Guarani can be found in http:// dx.doi.org/10.1016/j.quaint.2014.10.050; The authors of this study had no special access privileges others would not have.

**Funding:** This work was partly supported by the "Territorialidades Ameríndias no Alto Vale do Itajaí" financed in part by the "Elisabete Anderle Award 2017 - Segment material and immaterial heritage / archaeological research" (contract number 0148/ 2017 and process number 2578/2017) received by JSM, and is supported partly by the Coordenação de Aperfeiçoamento de Pessoal de Nível Superior - Brasil (CAPES, https://www.capes.gov.br/) - Finance Code 001, including APC doctoral fellowship. NP has received a productivity fellowship from CNPq (National Council for Scientific and Technological Development, http:// www.cnpq.br/)(CNPq 310443/2015-6). CL has received from CNPq a post-doctoral fellowship (CNPq 159440/2018-1).

**Competing interests:** The authors have declared that no competing interests exist.

species is how Grinnell [4] defined a niche. The development of the niche concept incorporated new factors: biotic interactions were considered [5], environment was expanded to multiple layers [6], and the effect of negative biotic interactions (e.g., competition) on species niche retraction were intensely evaluated [7–9]. Subsequent evolution of the niche concept recognized that positive interactions (e.g., facilitation) were capable of widening the niche [10]; introducing the idea that some species can modify their environment, and thereby can promote changes in resource availability for another species. These species are called ecosystem engineers [11]. The most effective ecosystem engineers are species that cause the longest-lasting modifications to their environment, and have the largest population size; for example, humans [11]. Once species modify their environment, they are able to change the direction and force of selection. This bi-directional relationship between species and their environment, may affect the species' niche as well as the niches of other species [12]. The Niche Construction Theory (NCT) predicts that past bi-directional selective processes result in species composition signatures that are perpetuated over time, and can be recognized in contemporary communities, a pattern called 'legacy' [12]. Different societies placed in different contexts modify their environments in different ways [1]. The term Cultural Niche Construction (CNC) has been coined to emphasize that cultural factors drive decisions about the changes that humans promote in their environment and, consequently, to the environment available to other species [13, 14]. CNC theory provides an integrative scenario to understand human legacies on natural ecosystems worldwide [14].

Recent historical, archaeological, and ethnographical studies recognize the historical influence of human activities on what has previously been considered primary, untouched, or pristine forests. Many tropical forests previously thought of as pristine were revealed to have been shaped by past human societies [15, 16]. For example, recent studies have shown that Amazonian forests have been modified by indigenous populations for millennia, altering plant species distributions across the region [17, 18].

The Atlantic Forest is likely no exception: the region was also occupied by Amerindians long before and continued after European arrival [19]. Archaeological studies have reconstructed the long-term human history of the region, identifying many Holocene archaeological sites in southern Brazil (at least 1704 sites already mapped in the region). This indicates that the Atlantic Forest, now drastically reduced and fragmented [20], has had a long history of human interaction.

History of human occupation in the State of Santa Catarina (Fig 1) commenced in the highlands ~ 11,500 Before Present (BP) [21] and in the littoral zone ~ 8, 000 BP [22, 23]. Hunters and gatherers were the first human groups to occupy this territory [24]. Current indigenous peoples in the area belong to the Southern-Jê (Xokleng or Laklaño and Kaingang) and Guarani linguistic groups. Jê groups from Central Brazil has started their migration to Southern around 3, 000 years BP [25] [26]. Archaeological dating in southern Brazil indicates The Southern-Jê commenced occupation along highland rivers, and then moved into littoral areas [25]. Highland occupancy is a general distribution pattern of Macro-Jê linguistic groups in Brazil in the central and eastern plateaus [26]. In the southern Brazil highlands, they relied on 'pinhão' (*Araucaria angustifolia* seed) as a key food source [27, 28], associated with another cultivated resources, as *Zea mays*, *Manihot esculenta*, and *Dioscorea* sp., and widespread hunting and fishing. The Guarani migration is related with Tupi expansion from Amazon. The main route for this Guarani migration from Amazon basin to southern Brazil is related to Paraná and La Plata river basin The two principal Guarani suggested centers are the Amazon and La Plata Basins [29]. The Guarani arrived in Southern Brazil only around 1800 years BP [30]. They are recognized as sealers, fishermen and farmers who followed the main rivers and the coast expansion of their territories. Contrasting both migration dynamics, Guarani groups migrated

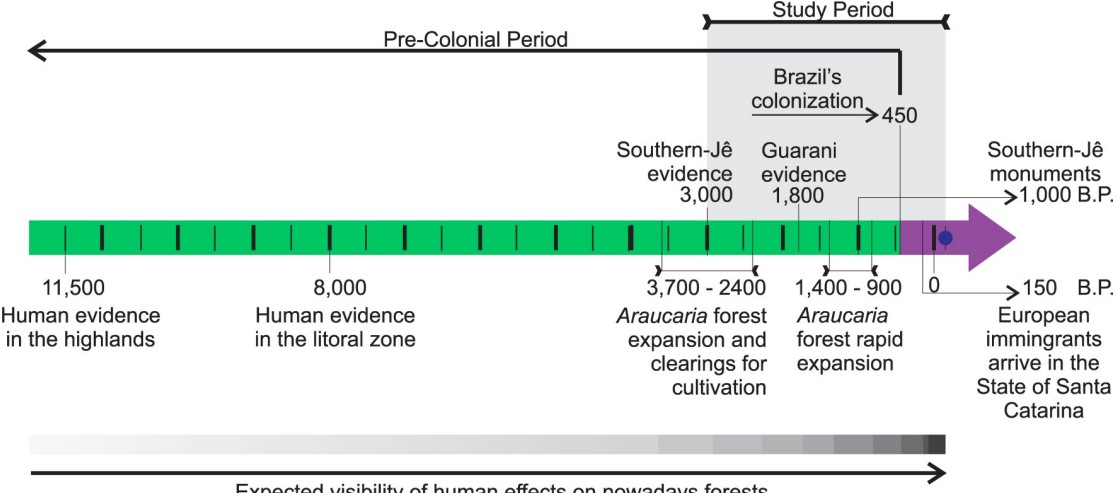

**Fig 1. Timeline of diachronical succession of human population events and linked changes in vegetation cover, in State of Santa Catarina.**

by expanding their territory, whereas the Southern-Jê migrated by leaving old territory and moving into new ones [25, 26].

The Southern-Jê and Guarani migrated to the south of Brazil to a landscape where forests were expanding and at a time of climate transition. In the highlands, *Araucaria* moist forests were expanding over what were previously grasslands. Grasslands are remnants of a drier climate, whereas forests are favored by the current climatic conditions [31]. Around 3000 years ago, forest expansion over grasslands intensified [32, 33], coinciding with the arrival of the Southern-Jê people in southern Brazil [25, 26]. Together with climate changes, humans may have acted as a complementary driver of forest expansion due to transportation of seeds [27, 28, 34, 35], and maintained grasslands using fire [32, 33, 36]. Recent research has shown that human action was essential for these forests to reach their maximum distribution [34, 35]. There are many evidence of the long-term use of *A. angustifolia* by past Amerindian societies also suggests they promoted the population expansion of this species, which is the dominant species of Araucaria moist forests in southern Brazil [37]. *Araucaria angustifolia* is recognized as a nurse plant, that is, its dispersion would favor the expansion of other forest species [38–40]. The role of human action in the assembly of forest communities in the pre-Colombian period as well as legacies in recent communities are still issues to be understood in the historical ecology of southern Brazil.

Globally, studies showing the influence of past cultural groups on species distribution patterns have intensified (see [16]); however, the long-term history of the Atlantic Forest is still poorly understood. Historical ecology studies in this region are hampered by the intensification of human occupation in recent times, which is obscuring past human legacies. This study aims to address this knowledge gap for southern Brazil. The niche modeling approach was used to understand the distribution of past Southern-Jê and Guarani Amerindian groups and then correlate this distribution with modern forest composition. Based on the premises that, (a) Southern Brazil has been occupied by humans for millennia, and (b) humans are cultural niche constructors, our hypothesis is that floristic composition differs in Southern Atlantic Forests with a high probability of past human activities, and that different cultural groups also leave differing floristic composition and abundance legacies notable yet, despite the disruption of these cultural groups and of the intense changes caused by recent populations. We suggest

that humans have been shaping these forest communities for at least 10,000 years; and that some differences in floristic composition are the results of the historical process of human occupancy and cultural variability.

## Materials and methods

### Study area

The study area (Fig 2) covers part of the pre-colonial Southern-Jê and Guarani distribution, in the Southern portion of the Neotropical Region. Located in southern Brazil, the State of Santa Catarina (SC) was selected because floristic data has been systematically collected across the state [41], providing the opportunity to compare spatial patterns with pre-colonial occupation. The entire area of SC is in the Atlantic Forest Domain [42], one of the global hotspots for biodiversity conservation [43]. Vegetation consists of coastal scrubland, mangroves, grasslands, and forests. Variation in forest types is influenced by geomorphology and climate. The entire State is in the subtropical zone and climatic variation is related to latitudinal and altitudinal gradients. The elevation gradient induces variation in air pressure, and consequently, promotes cooling. Landforms also drive atmospheric water movement and influence rainfall regimes [44]. Temperature variability increases with distance from the coast [44]. Santa Catarina has eighteen hydrographical basins, and combined with a moderate to high annual rainfall, has substantial river resources for local populations.

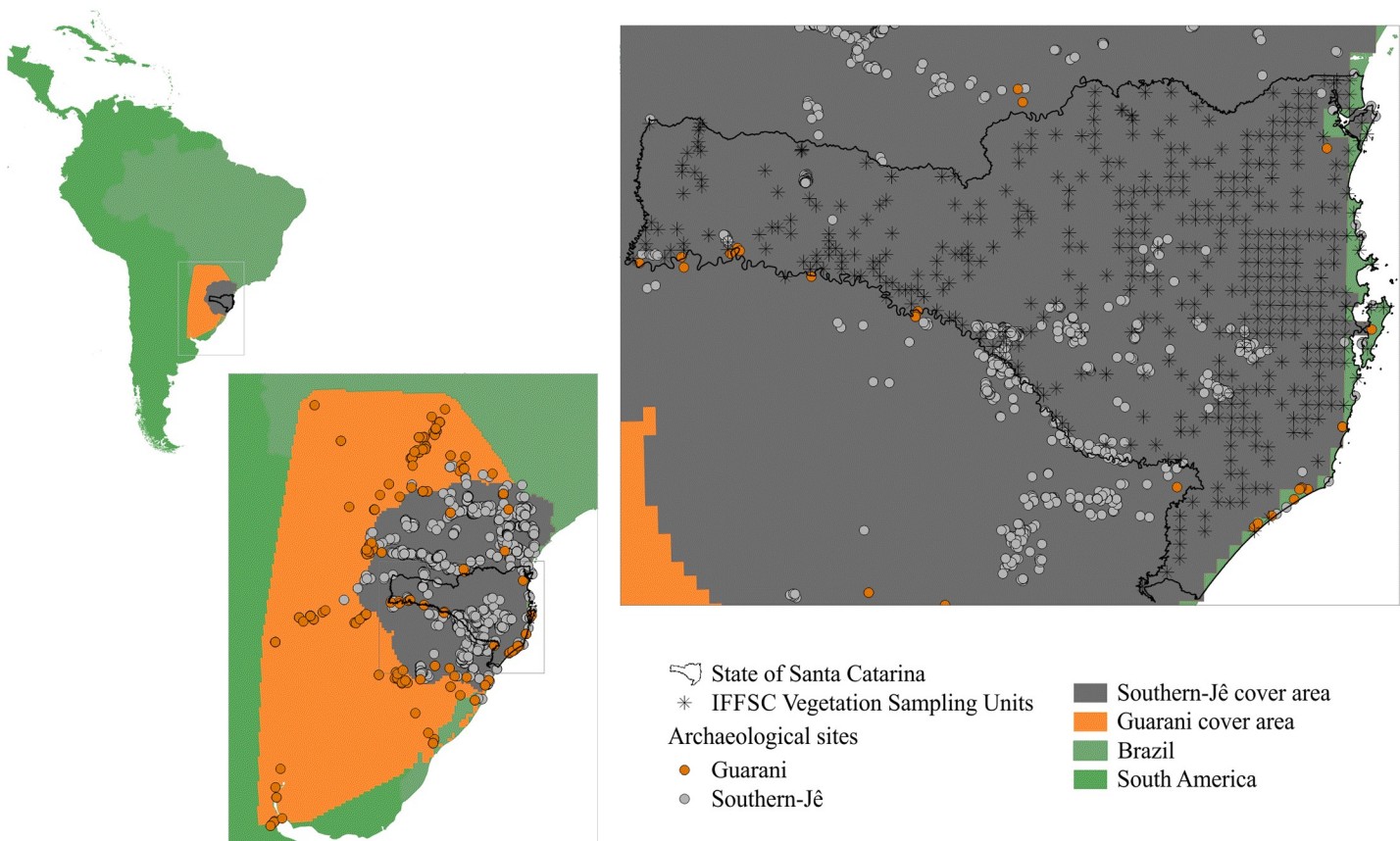

**Fig 2. Map of South America highlighting the study area that comprises the State of Santa Catarina.** Points indicate archaeological sites (Guarani sites in orange points and Southern-Jê sites in gray points) and stars are the floristic sample units.

## Vegetation data

We investigated whether there is a legacy of past human populations on present-day forest composition. Vegetation data used in this study was collected by the Santa Catarina Forest and Floristic Inventory (SCFFI), which systematically sampled shrubs and trees across the State in a grid system [41, 45] (Fig 1). Each sample unit covered 4000 m$^2$ [41, 45]. All forest plots except sand dunes and mangroves (a total of 417 sample units) were used to evaluate the abundance and distribution of all tree and shrub species in the study area.

## Archaeological data

Aiming to identify past human effects on forest composition, the first step was to locate archaeological sites. Archaeological sites within the study area were identified from literature reviews, as well as field data collected since 2011, by the Laboratory of Interdisciplinary Archaeology Studies at the Federal University of Santa Catarina. The literature review considered data published by Bonomo et al [30], and Noelli and Souza [25], covering the south of Brazil, and only used sites that were classified as Southern-Jê or Guarani. Another Southern-Jê archaeological sites were identified in LEIA database based on earthworks description. Using this data, we constructed a matrix with geographical coordinates of Southern-Jê and Guarani archaeological sites. Geographical location was the only information available for spatial analysis that could be obtained from all archaeological sites (although not always mentioned in archaeological studies). Only archaeological data with cultural attribution was used in order to identify any association between species and cultural variability. The selected archaeological sites have been occupied over the past 3,000 years.

## Topographic, hydrographic and forest type data

In order to understand which landscape features most influenced past Amerindian distribution, and contributed to defining their niche, we selected variables related to human preferences for specific environments; such as, topography, hydrology [46], and river proximity [23, 47]. The Height Above the Nearest Drainage (HAND) was used as a proxy for water table depth, and altitude and slope were used to define terrain (topography) [48]. Water course vectors were separated using Otto Pfafstetter's watershed coding method [49]. Coastline distance was used, assuming that the sea is a source of food and other services used by the populations [30, 47, 50]. Correlations between geomorphologic and hydrographic variables were evaluated. Forest types were mapped according to the classification of global terrestrial biomes [51] as follows: (1) Rain Forest (locally called "Dense Forest") on the coastal mountain range; (2) Mixed Forest (locally called "*Araucaria* Forest") on the highlands; and (3) Seasonal Forest found inland, especially in the Uruguay River basin [52].

## Data analysis

River classes area were grouped in case of collinearity (other variables did not present autocorrelation). The distance from each river class was rasterized using the Euclidean Distance function in Qis. We also generated a Euclidean raster distance from the coastline.

Occurrence area of each Amerindian group was defined based on the geographical distribution of Southern-Jê and Guarani archaeological sites. Occurrence area was used to limit the spatial area modeled for each cultural group. To understand the past Amerindian distribution, we developed models based on archaeological, topographic, and hydrographic data in Maxent interface (Ecological Niche Model–ENM; using the package 'ENMeval' [53]) to fit models and predictions of archaeological sites beyond sampled locations in R [53, 54]. Since we only had

presence-only data on the location of archeological sites and because data wasn't obtained by systematically sampling, we decided to use Maxent, Maximum Entropy Method, suited for analyzing presence-only data [55]. To further minimize sampling bias, we worked only with one archaeological site on every $10 \times 10$ km pixel. We used random k-fold validation with 4 k-folds, combining Features Class (FC): Linear (L), Linear and Quadratic (LQ), Hinge (H), and Linear Quadratic Hinge (LQH); and Regularization Multiplier (RM) sequence values 0.5, 4, 0.5. Models were sorted in decreasing order by the largest AUC values (Area under receiver operating Curve) and lowest overfitting, estimated by contrasting AUC values of train and test sets, and then, between top ranking models. Only one model was finally chosen by visual inspection. The chosen model was then used to generate predictive maps of the potential distribution of past Amerindian occupation. Estimates of the variables contribution to each model was expressed in terms of percent contribution and permutation importance values. Two separate maps for Southern-Jê and Guarani sites were constructed. Using these maps, the overlap of Southern-Jê and Guarani past distributions was calculated with the similarity statistic 'I' [56], where results fall between 0 and 1 (0 indicating no overlap and 1 indicating full overlap).

Vegetation sampling locations were overlaid with the maps of the potential distribution of both cultural groups. Next, we extracted the probability of each vegetation sample location falling over Southern-Jê or Guarani sites, or both. To assess the relationships between past Amerindian groups distributions and floristic composition of current forest fragments, we carried out a Redundancy Analysis (RDA), using the package 'vegan' [57] in R. The matrix of species abundances was used as the response variable and the probabilities of Southern-Jê and Guarani sites for each site were used as predictors. Forest type was added to ordination diagrams to aid interpretation but was not used as a predictor in the analysis. QGIS software was used for all geoprocessing procedures and R for multivariate analyses.

## Results

Guarani and Southern-Jê distributions were influenced by the distance to first or second order rivers (using Otto Pfafstetter's hierarchy, Fig 3). The importance of this variable was 42.2% and 40% (contribution and permutation importance values, respectively) in the Guarani model and 22% and 31.9% in the Southern-Jê model. Guarani distribution was also influenced by coastline distance (contribution: 36.9%; importance: 51.8%), which, conversely, was the environmental variable with the lowest influence over Southern-Jê distribution (contribution: 0.2%; permutation importance: 0.6%). Elevation was the environmental variable with the highest influence over Southern-Jê distribution (contribution: 39.6%; permutation importance: 27.1%), being irrelevant to Guarani distribution (contribution: 1.2%; permutation importance: 5.2%). The most suitable environments for Guarani people were those with proximity to the sea or rivers, while Southern-Jê sites were mostly found in elevated areas and near rivers. The final Guarani model (Fig 4A) was generated with FC = H and RM = 2, showing an AUC = 0.83. The final Southern-Jê model (Fig 4B) was generated with FC = LQP and RM = 4, and had an AUC = 0.71. Final images are available in S1 and S2 Files.

Niche overlap between past distributions of Southern-Jê and Guarani was noticeable ($I$ = 0.64; Fig 5). Overlapping areas are located along the major rivers (1st and 2nd order) and along main rivers within intermediate-sized basins (3rd order, Fig 4).

The first two axes of the Redundancy Analysis (RDA) explained 7% of the data variation (Fig 6) and show (i) directional segregation between the two cultural groups and (ii) the links between the distribution of these Amerindian groups and forest typologies. Ordination of sampling units (Fig 6A) suggest Southern-Jê sites relate with Mixed Forests and Guarani sites are more related with Rain Forests. Seasonal forests were likely used by both groups. The

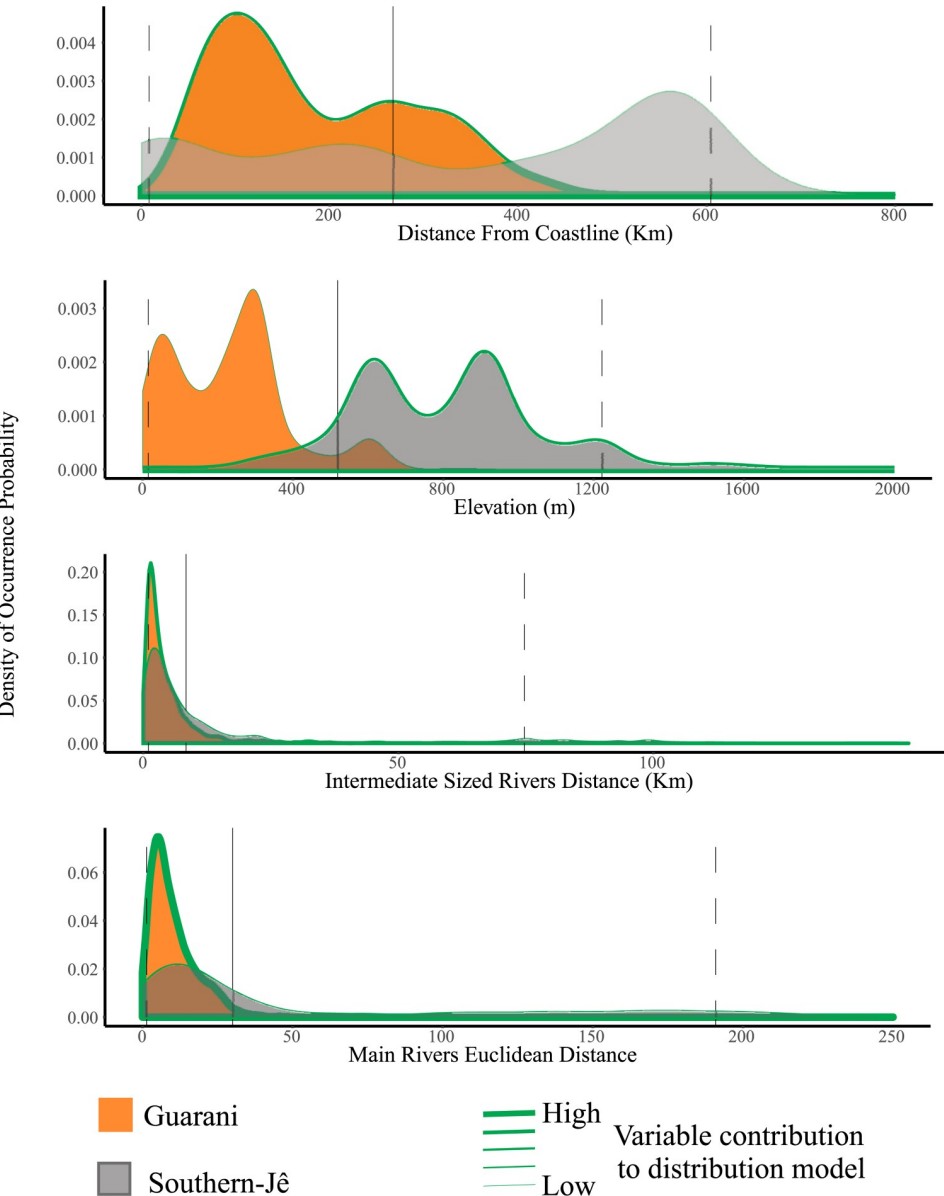

**Fig 3. Ridgeline plot of smoothed density estimates of the probability of occurrence of Southern-Jê (gray) and Guarani (orange) archaeological sites in relation to environmental variables.** Green line width indicates variable importance to the model.

scores of the relation between the first axis of the RDA and the past Amerindian groups distribution was -0.57 for Guarani and 0.85 for Southern-Jê: the second axis had positive scores for both groups (Southern-Jê = 0.52 and Guarani = 0.81). RDA ordination analysis highlighted three clear groups (Fig 6): (1) species shared between the Southern-Jê and Guarani groups that are in the positive range of the second axis, (2) species related to the occurrence of the Southern-Jê group in the positive range of the first axis, and (3) species related to the Guarani presence in the negative range of both axes. Some species related to both axes, but were more strongly related to the Southern-Jê (left side of the first RDA axis) or Guarani (right side of the first RDA axis). Twenty-nine species had ordination scores higher than 0.1 or lower than –0.1

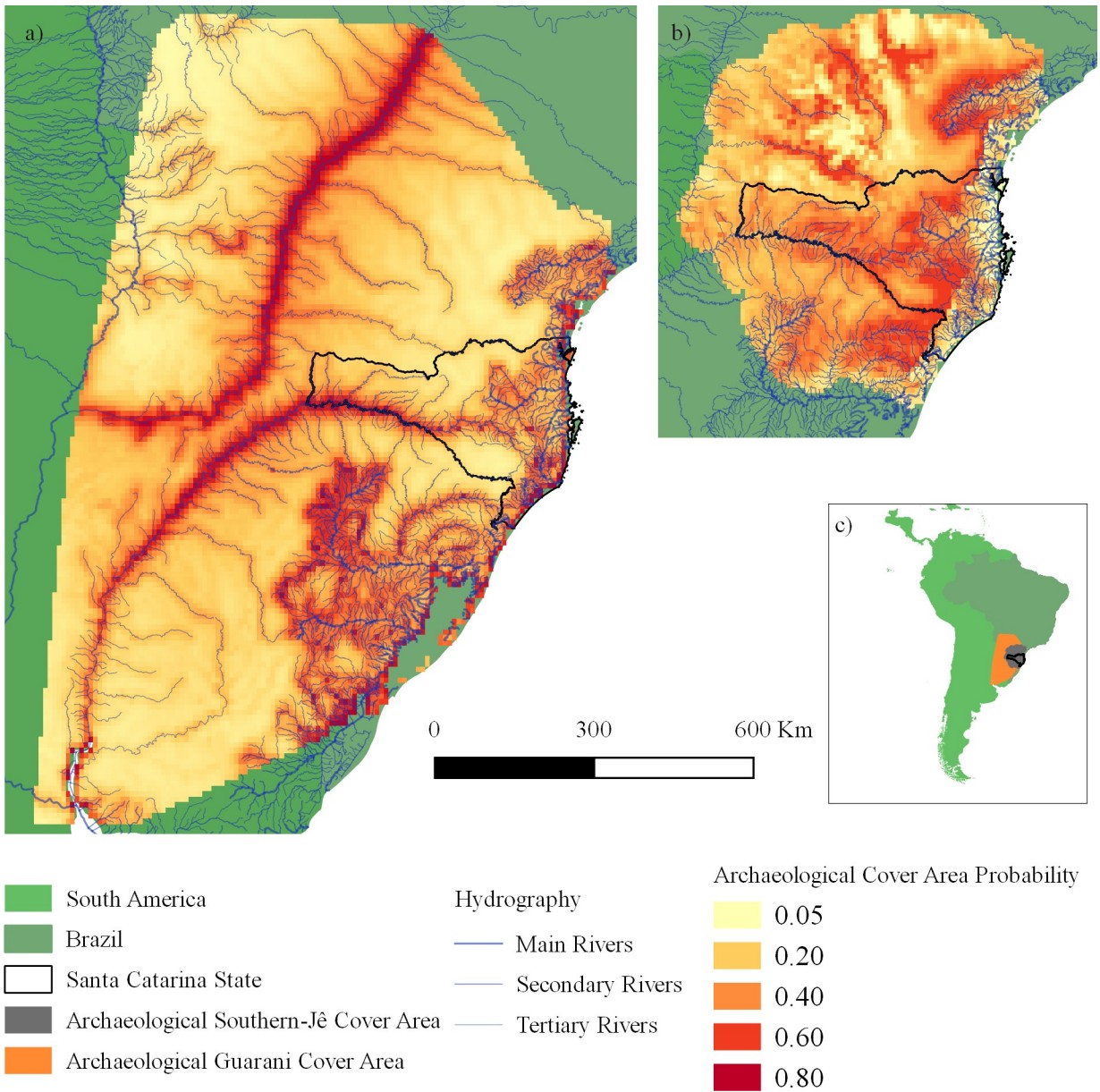

**Fig 4.** a) Map of the probability of past Guarani distribution; b) Map of the probability of past Southern-Jê distribution; c) Map without scale of South America highlighted areas showed in *a* and *b*. These maps were created with ecological niche models that presented the lowest overfitting and the highest AUC values.

and were considered as associated with Amerindian past distribution. These 29 species were listed along a cultural gradient (Fig 7). The listed species belong to 20 families, Lauraceae (n = 6), Sapindaceae (n = 3), Anacardiaceae and Arecaceae (n = 2), with the other families represented by only one species each.

## Discussion

This study investigated the cultural legacy of pre-colonial Guarani and Southern-Jê in the present floristic composition of forest remnants. Correlation of the distribution of past

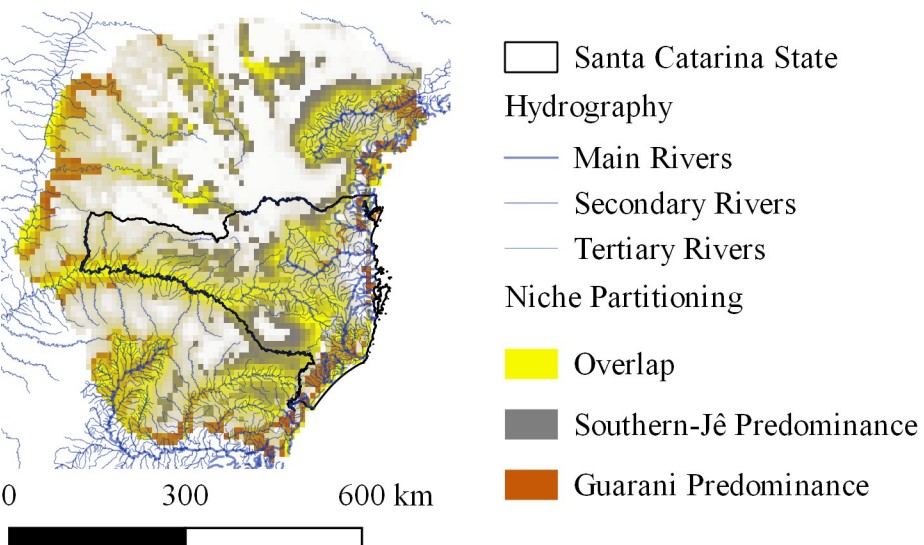

**Fig 5. Spatial partitioning between archaeological Southern-Jê and Guarani distribution.**

Amerindian groups and forest species detected species associated with each group. Landscape compartmentalization between the geographical distribution of archaeological sites occupied

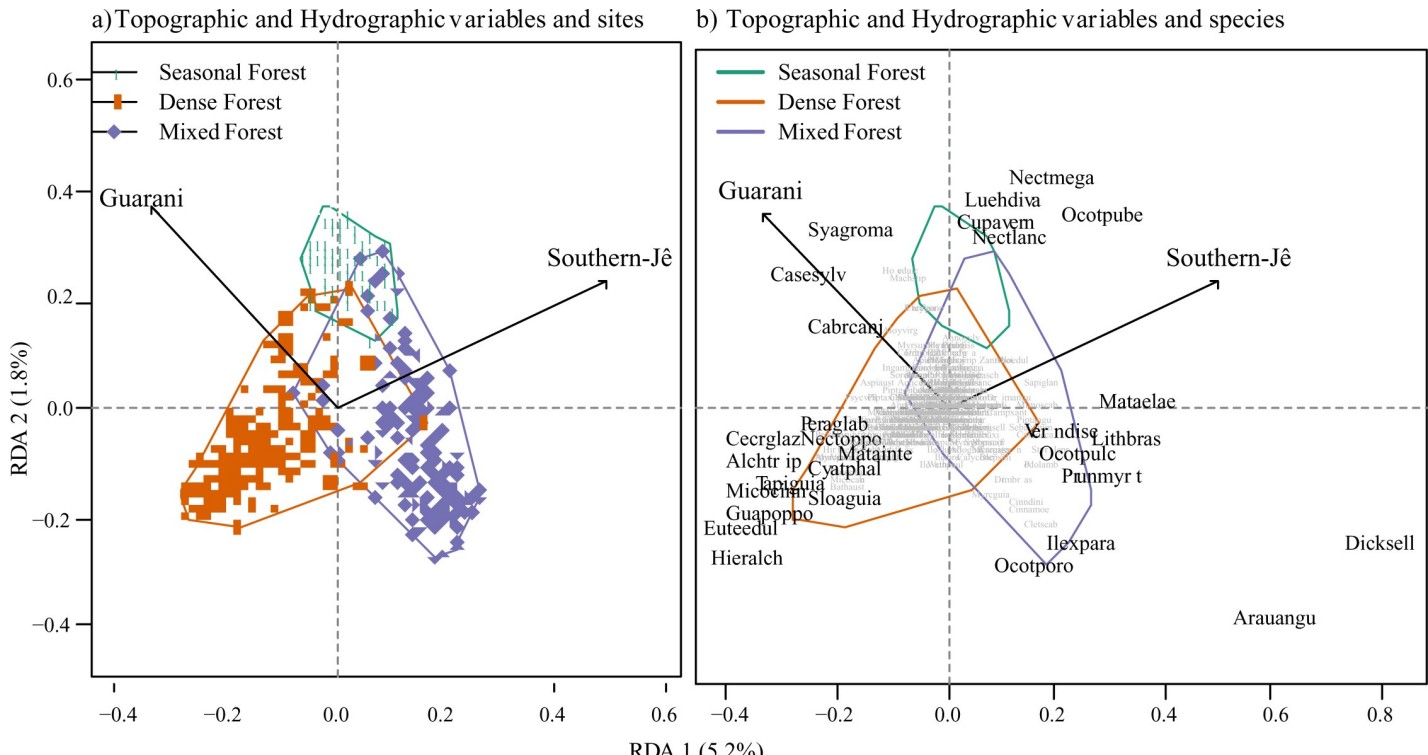

**Fig 6. The redundancy analysis of species abundance and the probability of occurrence of Southern-Jê and Guarani archaeological sites.** Arrows represent maximum variation directions. a) Dots represent vegetation sampling units. b) Shows the species distribution. Polygons and colors indicate forest typologies in both figures.

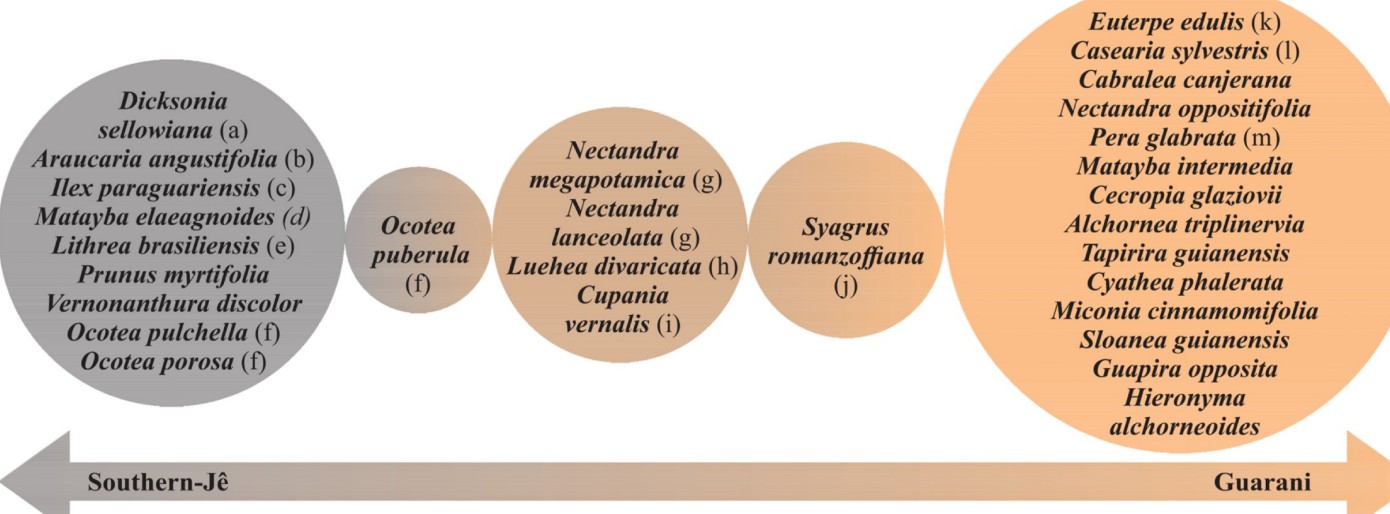

**Fig 7. Relationship between tree species and the two past Amerindian groups (Guaraní and Southern-Jê).** These species were selected because they had ordination scores higher than 0.1 or lower than −0.1 in the RDA.

by Southern-Jê and Guarani groups was identified. Landscapes suitable for Southern-Jê are generally at higher altitudes and close to water courses. Guarani settlements were more likely to be closer to the sea or large rivers. Suitable landscapes for both groups seem to overlap near the larger rivers.

The species that showed association with pre-colonial Amerindian groups could simply be associated with the landscape characteristics that predicted the distribution of these groups. Alternatively, the plant species could be attracted to human occupation or could be favored by humans, by both intentional or unintentional actions [58]. For instance, Amerindians intentionally cleared the land and used fire to produce crops, and unintentionally favored plants by interrupting seeds dormancy, like *Mimosa scabrella* [59–61], by improvement in light availability to pioneer species, like *Solanum mauritianum* [60]. In the State of Santa Catarina palynological evidence indicates that the opening of clearings for cultivation started between 3760 and 2430 B.P (Fig 1) [34]. Guarani expanded their occupation area and transplanted useful plants when they migrated, favoring fruit trees that attract fauna, and expanding the territories of species of cultural importance [62]. Amerindians transplanted cassava, beans, maize, and genipap from the Amazon to southern Brazil as part of a cyclic agroforestry production system [62]. The Southern-Jê were probably responsible for *Araucaria* forest expansion over grasslands [27, 34, 35]. Pre-colonial Southern-Jê and Guarani archaeology reveals evidence of agriculture [30, 62, 63] and pottery production [30, 62–64], and both these activities require fire. Fire was used for many things (e.g., heating, cooking, ceramic preparation, and cremation) and may have involved selective extraction or gathering of species for fuel. Fuel for funeral use must reach very high temperatures, for domestic use it must be easily controllable [65]. The use of fire in rituals can also lead to the selection of woods by symbolic characteristics. Archaeological research has shown that Southern-Jê used preferred species in contexts that indicate the practice of rituals: *Araucaria angustifolia*, and *Jacaranda* sp. in a location where Araucária is not available [66]. Fire use could also be unintentionally selective, driving changes in species composition, unintended by those lighting the fires. Agriculture and transplanting fruit trees are examples of intentional activities that modify species composition. Southern-Jê constructed earthworks, that were

landscape modifications benefiting some species, in a similar way to the Amazon mounds earth-works [17].

We observed a distribution pattern of Southern-Jê and Guarani groups that was associated with forest types. Southern-Jê sites occur mainly in Mixed Forests (*Araucaria* forests), where many subterranean structures, earthworks, have been found. *Araucaria* forests are recognized as an important element in the social organization of the current Southern-Jê (Kaingang and Xokleng/Laklaño) [34, 66]. Southern-Jê controlled their territories by managing the *Araucaria* forests [64]. Guarani sites have not been found in Mixed Forests but are common in Rain Forest areas. Both groups co-occurred in Seasonal Forests, probably because this forest type is mainly distributed along an important river (the Uruguay River). Rivers are a factor associated with both groups' distributions, including their ancestral territories. Rivers are especially important in Guarani culture, they are excellent sealers and fishermen [30]. The Southern-Jê arrived before the Guarani in southern Brazil and initially occupied coastal zones and riversides [25]. However, main rivers and the coastline were also migration routes used by the Guarani [30]. The Southern-Jê subsequently intensified the highlands occupation, and this may be due to arrival of the Guarani people [25, 67], or due to ecological identity with their ancestral territories, which are highland headwaters [26].

We found a species partitioning gradient between the two cultural groups in analysis of the species distribution (Fig 7). An important point is that our analyzes do not isolate the causes of the distribution of plant species, we are considering niche construction as a process in which factors of different natures are inseparable. Many of the species listed in this gradient are cited in the ethnobotanical literature as important for Southern-Jê and Guarani. Species such *Dicksonia sellowiana* (Fig 7A) is associated with Southern-Jê groups, and cited in literature as a cultur-ally valued species that is used to prepare a traditional drink called "Mõg" [68, 69]. *Araucaria angustifolia* (Fig 7B) appears to be associated with Southern-Jê presence, and archaeological evi-dence exists of its use by pre-colonial Southern-Jê groups [27, 70, 71]. *Dicksonia sellowiana* and *A. angustifolia* are very abundant species in Mixed Forests, as are *Ilex paraguariensis* (Fig 7C) and *Matayba elaeagnoides* (Fig 7D). *I. paraguariensis* is a key cultural species, whose leaves are used to prepare a drink called "chimarrão" or "mate", that is popular in South America [72]. The practice of consumption and processing of *I. paraguariensis* leaves are culturally tran; smitted by Amerindians [72]. *M. elaeagnoides* is described as a good quality burning wood [73]. *Lithraea brasiliensis* (Fig 7E) has a popular name "bugreiro" ("bugre" with an added suffix to indicate that it is connected to, or used by, the bugre). Although this species is associated with Southern-Jê, *L. brasiliensis* was also reportedly used by Guarani for construction purposes [74]. Along the entire gradient, we observed species of the Lauraceae family, composed by the genera *Ocotea* (Fig 6F) and *Nectandra* (Fig 7G). Among these and linked with Southern-Jê distribution, the species *Ocotea porosa* is currently listed as vulnerable to extinction [75]. We observed a change in the composition of Lauraceae inside the gradient. From the species are shared by both Southern-Jê and Guarani, the genus *Ocotea* is replaced by the genus *Nectandra*. The spe-cies shared by the two cultural groups are at the center of the gradient and all of them are abun-dant in inland Seasonal forests. In addition, all of these shared species are cited by various cultural groups in ethnobotanical literature, reinforcing that they are species of shared cultural use and knowledge. *Luehea divaricata* (Fig 7H) is used medicinally by Laklaño (Southern-Jê) [69], Guarani [74, 76, 77], and regional farmers [73]. *Cupania vernalis* (Fig 7I) is used by Lak-laño [69], Guarani [74, 76, 77] and regional nowadays farmers [73] for food, construction and artifact production. Species such as *Syagrus romanzoffiana* (Fig 7J) *and Euterpe edulis* (Fig 7K) are associated with Guarani presence, but only *S. romanzoffiana* was common to both Guarani and Southern-Jê. Both species are reportedly used by the Guarani [74] and *S. romanzoffiana* is known to be cultivated by them [62]. Arecaceae is an important family for human subsistence

in the Atlantic Forests, because its species produce many fruits, and in some cases, palm heart [19]. Arecaceae phytoliths have been found in archaeological Southern-Jê pottery, which could suggest consumption and processing or may represent background vegetation [63]. *Euterpe edulis* is a very abundant species in dense forest (Rain Forest) [78], while *S. romanzoffiana* also occurs in other forest typologies [78–80]. Other species associated with Guarani presence are *Casearia sylvestris* (Fig 7L) and *Pera glabrata* (Fig 7M), both species with Amerindian references in their popular name: "bugre tea" [73, 74, 76] and "bugre heart" [69] respectively ("bugre" is a pejorative name for native people). *Casearia sylvestris* is a species culturally prominent for present day Guarani [76] and *P. glabrata* wood is used by Guarani for construction [74, 77] and by Laklaños (Southern-Jê) for artifact production [69].

Our analysis explained only 7% of data variation; however, given the wide spectrum of environmental characteristics that may influence vegetation patterns at the landscape scale, past Amerindian cultures may be an important factor to consider. Our results indicate past Southern-Jê and Guarani distributions are another factor driving differences in present-day forest species composition.

To infer how much of the activities in the pre-colonial period may be registered in today's forest composition, it is important to consider demographic changes. We illustrate the expectation of human impact at different times (Fig 1). Although we still don't have detailed demographic estimates for the pre-colonial period in southern Brazil [24], we have estimate up 3,000 years ago [81]. The population density in the state of Santa Catarina would have been 0,15 inhab/km2 in the period between 11,000 and 7,000 years ago [81]. That density would have remained in the western portion until 3,000 years ago, but would have increased in the east reaching up to 0,75 inhab / km2 [81]. that we don't have demographic data, but in the period between 1,800 BP and 1,400 BP the Guarani occupation began [30], there was an increase in the Southern-Jê population [34], and as a consequence of this interaction there were cultural changes [67]. In the present year, 2,020, the population density in the state of Santa Catarina varies from 0.7 inhab/km2 (in the largest portion of the territory) to 400/inhab/km2 (in a few places) [82]. Thus, despite the collapse of the Amerindian populations, there was a population increase in the study area. That pattern contrasts with the Amazon, where there was general depopulation [58]. If on the one hand in the Amazon, populations of domesticated plant species are being lost [58], on the other hand, pre-colonial legacies are evident [18]. In State of Santa Catarina Amerindian populations have drastically reduced, just like in the Amazon, so populations of domesticated species may also be missing. However, if there was intercultural transmission of knowledge, some populations of domesticated species may have perpetuated. In general way, State of Santa Catarina population density increased and that has resulted in intense recent changes in vegetation [83]. Despite this, we have listed here 29 species related to the Southern-Jê and Guarani occupation in the pre-Colonial period. We suggest that traits of domestication of these species be investigated, because some populations of domesticated species in the pre-Colonial period may have been maintained [84], and we need more investigations in this regard.

## Past Amerindian distribution

This study found that Southern-Jê and Guarani group distributions are related to different environments. Landscapes suitable for Southern-Jê were likely higher altitude and close to minor water courses, whereas Guarani settlements were more likely located near the sea. Both groups may have shared locations near large rivers.

The past distribution models generated, are an advance on broad analyses, because they reinforce, and spatialize the preferential landscape characteristics for pre-colonial Southern-Jê

and Guarani groups. Maps of past Amerindian group distributions are useful for investigating spatial patterns; such as, relationships with vegetation, or landscape partitioning between cultural groups. Distribution models generated with presence data are only potentially usable when they have an AUC value above 0.75 [85]: however, for goodness of fit evaluation of models in general, AUC values above 0.7 can be considered good [86]. Evaluating AUC values in species distribution models, Elith et al (2006, [87]) observed that 64% of the best models presented AUC values above 0.75, and 14% of the best models presented AUC values between 0.7 and 0.75. In this respect, the Guarani distribution model, with a value of 0.83, can be considered adequate. The Southern-Jê model presented an AUC above 0.7, so we consider it acceptable, but understand there are caveats. Our maps explain the spatial dimension of the occupation of southern-Jê and Guarani, but the temporal dimension mixes the entire period of pre-Colonial occupation.

There was a niche overlap between both cultural groups near main rivers. This may be explained by occupation at different times, or by simultaneous occupation involving interactions between the two groups and remains to be further addressed. The Southern-Jê initially occupied diverse territories in Southern Brazil. Around the time the Guarani arrived and began to occupy areas close to large rivers and the coast the Southern-Jê were settled in the highlands. Although there are records of Southern-Jê presence in the highlands since 2000 years BP, it is only around 1000 years BP that these groups started to build their funerary earthworks [64, 67]. This new architectural expression may be a way of controlling their territory, because it coincides with the arrival of Guarani groups [67]. Guarani groups expanded their occupation along the margins of major rivers and along the coastline [30]. The Jê people were distributed over the highlands in their Brazilian territory [25, 26], and their pattern of migration created isolated groups, differing from the Guarani [25]. Following contact with Europeans, Amerindian populations were reduced due to territorial conflicts, diseases, political, and social disruption. There is evidence of Guarani occupation followed by subsequent Southern-Jê occupation in some locations [25], suggesting that the groups competed for the territories. Furthermore, synchronous Guarani sites inside of Southern-Jê areas have been found, suggesting fluid frontiers where these groups have interacted over time [25].

## Conclusion

Although there are still many pieces of the puzzle to be put together for a complete understanding of vegetation patterns, this study is the first to assess pre-colonial cultural effects, on the contemporary forest composition, of a large extent of the Atlantic Forest. This study evaluated forests that have been seriously altered post-European colonization [83] and still detected a legacy of pre-colonial cultural groups on forest composition. Forest composition differs in landscapes previously occupied by different Amerindian groups. This difference may be due to the different ways differing cultures manage territory, indicative of Cultural Niche Construction. We conclude that long-term cultural activities may have acted with other biotic and abiotic processes to determine forest compositions in the Southern Atlantic Forest.

## Supporting information

**S1 Code. R Scripts.** This file contains all the codes wrote in R.
(DOCX)

**S1 File. ENM Guarani raster.** This file can be opened using a Geographic Information System (Gis Software) or following the script provided in S1 Code.
(CSV)

**S2 File. ENM Southern-Jê raster.** This file can be opened using a Geographic Information System (Gis Software) or following the script provided in S1 Code.
(CSV)

**S3 File.**
(CSV)

**S4 File.**
(CSV)

**S5 File.**
(CSV)

**S1 Data.**
(RAR)

**S2 Data.**
(RAR)

**S1 Fig.**
(TIF)

## Acknowledgments

We thanks to Francisco Noelli and Jonas Gregorio de Souza, for Southern-Jê data. We are also grateful to the Laboratory of Interdisciplinary Studies in Archaeology (LEIA) at the Federal University of Santa Catarina (UFSC) for archaeological data; thanks to Leonardo de Campos who made many helpful suggestions on an earlier draft; thanks to Mário Tagliari and Ramiro Melinski for contributions in ecological niche modeling; thanks to Christopher Graves for English revision; and thanks to Thiago Ehlert for the tabulation of useful plants data.

## Author Contributions

**Conceptualization:** Aline Pereira Cruz, Juliana Salles Machado, Lucas Bueno, Nivaldo Peroni.

**Data curation:** Aline Pereira Cruz, Carolina Levis, Juliana Salles Machado, Lucas Bueno, Nivaldo Peroni.

**Formal analysis:** Aline Pereira Cruz, Eduardo Luiz Hettwer Giehl, Nivaldo Peroni.

**Funding acquisition:** Juliana Salles Machado, Nivaldo Peroni.

**Investigation:** Aline Pereira Cruz, Eduardo Luiz Hettwer Giehl, Nivaldo Peroni.

**Methodology:** Aline Pereira Cruz, Eduardo Luiz Hettwer Giehl, Carolina Levis, Lucas Bueno, Nivaldo Peroni.

**Project administration:** Juliana Salles Machado, Lucas Bueno, Nivaldo Peroni.

**Resources:** Juliana Salles Machado, Nivaldo Peroni.

**Software:** Aline Pereira Cruz, Eduardo Luiz Hettwer Giehl.

**Supervision:** Lucas Bueno, Nivaldo Peroni.

**Validation:** Aline Pereira Cruz.

**Visualization:** Aline Pereira Cruz.

**Writing – original draft:** Aline Pereira Cruz, Eduardo Luiz Hettwer Giehl, Carolina Levis, Nivaldo Peroni.

**Writing – review & editing:** Aline Pereira Cruz, Nivaldo Peroni.

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
