## [Decision Letter · Decision Letter 0]

15 Jan 2020

PONE-D-19-33906

Pre-colonial Amerindian legacies in forest composition of southern Brazil

PLOS ONE

Dear Dr. Pereira Cruz,

Thank you for submitting your manuscript to PLOS ONE. After careful consideration, we feel that it has merit but does not fully meet PLOS ONE’s publication criteria as it currently stands. Therefore, we invite you to submit a revised version of the manuscript that addresses the points raised during the review process.

We would appreciate receiving your revised manuscript by Feb 29 2020 11:59PM. To enhance the reproducibility of your results, we recommend that if applicable you deposit your laboratory protocols in protocols.io, where a protocol can be assigned its own identifier (DOI) such that it can be cited independently in the future. For instructions see: http://journals.plos.org/plosone/s/submission-guidelines#loc-laboratory-protocols

We look forward to receiving your revised manuscript.

Kind regards,

André Jasper, Ph.D.

Academic Editor

PLOS ONE

Additional Editor Comments:

Dear Dr. Pereira-Cruz,

After carefully read you manuscript and both reviews opinion, I’m sending it back to you with some comments to be considered and included, if it seems relevant to you.

Both reviewers agree that this is a very interesting and original manuscript that should ultimately be published in Plos One.

Although the reviews are not conflicting, I agree with Rev #1 that a few issues must be attended before the manuscript can be accepted.

So, I kindly ask that you consider the suggestions made by Rev. #1 and do some effort to attend it.

If you do not agree with any of the suggestions made, you must justify it in accordance to Plos One editorial managing policies.

I’m looking forward to getting your revised version as soon as possible so that we can proceed with the evaluation process.

With kind regards,

André

3. We note that Figures 1, 3 and 4 in your submission contain map images which may be copyrighted.

a. You may seek permission from the original copyright holder of Figures 1, 3 and  to publish the content specifically under the CC BY 4.0 license. 

Reviewers' comments:

Reviewer's Responses to Questions

**Comments to the Author**

1. Is the manuscript technically sound, and do the data support the conclusions?

Reviewer #1: Partly

Reviewer #2: Yes

2. Has the statistical analysis been performed appropriately and rigorously? 

Reviewer #1: I Don't Know

Reviewer #2: N/A

3. Have the authors made all data underlying the findings in their manuscript fully available?

Reviewer #1: Yes

Reviewer #2: Yes

4. Is the manuscript presented in an intelligible fashion and written in standard English?

Reviewer #1: Yes

Reviewer #2: Yes

5. Review Comments to the Author

Reviewer #1: This is a very interesting and original research on historical ecology carried out in Santa Catarina State in Brazil. The paper is fairly well organised, the topic is very important and deserves to be published. The Introduction is well-written and interesting.

However, in other part the text is sometimes not so clear and there are paragraphs that should be read again to become easy understandable.

As a general advice, I suggest to consider some aspects to be incorporated in the paper:

a) The chronological framework and diachronical succession of events seems to me not discussed; please, consider that the international audience can be not informed about what means ‘pre-colonial’ – and when the change from the hunter-gatherer to Neolithic culture (transitions or migrations?) had occurred in Brazilian regions.

b) The demography datum is missed. One should consider that the number of people changed during the different chrono-cultural phases, and therefore when you mention 11,500 BP hunter-gatherers, were they fewer or had a similar impact than the more recent human groups?

c) Are there known differences in the cultural choices (environmental-dependent?) of the human groups involved in the research? (probably they are evident from the Fig. 6 ) Or the differences are also linked to the changes of one culture along the changing millennia?

Line 36 = Araucaria can be added to the keywords

Lines 68-69 = how long is this ‘long-term’ history? It emerges from the next paragraph that it is about 11,500 years long. Therefore, a more precise chronology can be added, such as changing “identifying many archaeological sites” to “identifying many Holocene archaeological sites”, or something similar.

Line 103 = “humans are cultural niche constructors” = this statement is absolutely sharable and there is a general consensus on this vision, as well as “different cultural groups also leave differing floristic composition legacies” = please note, the presence of archaeological sites and archaeological/archaeobotanical findings is mandatory to infer the type of land use. Further information can be obtained from the ethnobotanical review, as in the discussion of this paper.

Line 244 = the first part of the discussion is well centred on geographical and geomorphological characteristics that can explain the observed results and trends. However, the timeline variable – chronology of archaeological sites/events is not reported in the following paragraphs.

Line 253 = ‘Alternatively, the species could’ = Alternatively, the plant species could

Line 255 = as an example on the need of chronological details, in the sentence “Amerindians intentionally cleared the land and used fire to produce crops,”, you should add the culture and/or Holocene period or millennia BP.

Line 264 = “may have involved selective extraction of species for fuel” = is this a speculative argument or can you make an example from a case study?

Lines 279-310 = This is an interesting ethnobotanical part of the text but it is not clear in which order the plants/uses are discussed, and why they are listed.

Line 285 = phytolits in pottery could suggest consumption or, preferably, are they evidence of plant processing?

Line 385 = practical

Citations 31. 32, 33, 77 = put ‘Aracauria’ in italics

Citation 37, 55 = check the format

Fig. 6 = it is an interesting figure (please, quote it in the text) but I am not sure that it is sufficiently clear the sense and aim of this ‘separation of plant species’ – Cultural? Phytogeographical? Current or past different exploitation? The full name of species should be preferred to the abbreviation of the genus, if possible.

Reviewer #2: The article PONE-D-19-33906 “Pre-colonial Amerindian legacies in forest composition of southern Brazil” presents the analyzes on patterns of occupations and settlements of ancient societies in Brazilian lands. The evidence left by Amerindian societies (especially Southern-Jê and Guarani) is persistent in forests around the world. This research presents some important results about how these societies have affected and managed over time the forest structure in southern Brazil. The investigation allows to evaluate the tree composition of forest fragments and the corresponding environmental variables in the state of Santa Catarina. Archaeological data were used to understand ecological niche models thinking of prediction models. The amount of inventory studied is quite significant. The methodology used is correct and can be an excellent tool for further studies. It is noteworthy that the settlements and sites of these follow the pattern already studied since the 1960s by American and Brazilian archaeologists. Which serves as confirmation of some procedural models. As for the results and analyzes on the floristic composition of forest remnants and the relationship of the distribution of ancient Amerindian groups and the species associated with each group, the sites demonstrate the patterns of occupation at altitudes and reliefs already highlighted in other classical studies of Brazilian archaeology. Structurally the manuscript is in accordance with a scientific text following the ethical and methodological rigors. The bibliography is up to date and of excellent quality. The manuscript deserves to be published and will bring important contributions to current studies on ecological and environmental history in southern Brazil, still ephemeral in archaeological studies.

6. PLOS authors have the option to publish the peer review history of their article (what does this mean?). If published, this will include your full peer review and any attached files.

Reviewer #1: No

Reviewer #2: No

---

## [Author Response · Author response to Decision Letter 0]

29 Feb 2020

Dear Editors,

We are grateful to you and reviewers for comments regarding our work. The comments were very positive and contributed to improving the manuscript.

In short, we are sharing several files, including all the code we have written, to improve the reproducibility of our analyzes. Changes were made in the manuscript and figures to address the issues raised in review process. 

We hope that with the improvements in the manuscript it can be published in Plos One.

RESPONSE TO JOURNAL RQUIREMENTS

Journal Point 1: “Please ensure that your manuscript meets PLOS ONE's style requirements, including those for file naming.”

Response to Journal Point 1: We check and adjust the style requirements in the manuscript and in the filenames.

Journal Point 2: “We note that you have indicated that data from this study are available upon request. PLOS only allows data to be available upon request if there are legal or ethical restrictions on sharing data publicly.”

Response to Journal Point 2: We organize the data in order to preserve the third-party data, but to allow the analyzes to be reproduced. All steps of the analysis in R, containing the names of the files we are sharing and / or the download paths are in the file “Methods.docx”. 

We are sharing two georeferenced images, with the results of niche modeling. The Guarani model is available in the archive “S1A_result.tif” and the Southern-Jê model in file “S1B_result.tif”.

Floristic data are third-party Data. We are sharing the abundance matrix with anonymization of species names and sample units. These data belong to the Floristic Forest Inventory of the State of Santa Catarina (IFFSC), and are published in books, available for download at: 

<https://sites.google.com/view/iffportal/publica%C3%A7%C3%B5es/livros>. 

We requested the abundance matrix (raw data) from the IFFSC coordination, through an email with a description of our research objective. The project coordinator is Dr. Alexander Christian Vibrans, from the Regional University of Blumenau - FURB. Your emails are:acv@furb.br; iffsc@furb.br; inventarioflorestal.sc@gmail.com. 

We are sharing part of the geographical coordinates of the Southern-Jê archaeological sites in the named file “S1B_POINTS.csv”. We do not share the portion of the data that is published in Noelli e Souza, 2017, (doi: 10.1590/1981.81222017000100004) and was provided by the authors (thirdy-part data). To request this data, the authors must be contacted by e-mails: 

Jonas Jonas Gregorio de Souza <jonas.gregorio@yahoo.com.br>; 

Francisco Silva Noelli <chico.noelli@gmail.com>.

Journal Point 3: “We note that Figures 1, 3 and 4 in your submission contain map images which may be copyrighted.”

Response Journal Point 3: All figures are original and was created by authors. We don’t utilize proprietary data.

RESPONSE TO REVIEWERS

We thank the reviewers for the assessment and comments of our manuscript. We have modified the text and figures according to the referees’ critiques. Below are our comments on each issue raised by Reviewer #1.

Reviewer Point 1: “This is a very interesting and original research on historical ecology carried out in Santa Catarina State in Brazil. The paper is fairly well organized, the topic is very important and deserves to be published. The Introduction is well-written and interesting. However, in other part the text is sometimes not so clear and there are paragraphs that should be read again to become easy understandable.”

Response to Reviewer Point 1: We hope that the text is clearer with the changes. If there is any critical point, please do not hesitate to let us know.

Reviewer Point 2: “As a general advice, I suggest to consider some aspects to be incorporated in the paper: a) The chronological framework and diachronical succession of events seems to me not discussed; please, consider that the international audience can be not informed about what means ‘pre-colonial’ – and when the change from the hunter-gatherer to Neolithic culture (transitions or migrations?) had occurred in Brazilian regions.”

Response to Reviewer Point 2: We agree with the reviewer on this important point about succession of events. We included a timeline (Fig 1, in line 89), with the aim of making the sequence of events in the population of the State of Santa Catarina better understandable.

About culture transitions, research conducted in recent years have pointed to the complexity of the subject due to regional variations, linked to specific cultural histories and the unsuitability of a consolidated basis for the study of European prehistory in the tropics. In general, these studies have shown that the aspects traditionally considered as key elements for this “Neolithic transition”, such as ceramics, sedentary lifestyle, agriculture, do not appear together at once time. There are several examples in tropical regions for a very old presence of ceramics, which, however, does not remain and is constant or in continuous growth, involving, in some regions, production cycles and disappearance. At the same time, there are indications of domesticated items in contexts with dates for the Early Holocene, which are not associated with the presence of ceramics or changes in the pattern of mobility. In other words, these aspects integrate local historical trajectories, with regional variations in chronological and organizational terms. In southern Brazil, what we have identified is a change in the distribution pattern of the sites and their associated archaeological remains from the 3,000 years BP, when a process of cultural interaction is intensified associated with the migration of Jê groups to the southern plateau. However, on the coast, since 7,000 years ago, with intensification between 5-3,000, significant changes involving demography and territoriality are identified and associated with the construction of large shell mounds.

Reviewer Point 3: “(…) I suggest to consider some aspects to be incorporated in the paper: (…) b) The demography datum is missed. One should consider that the number of people changed during the different chrono-cultural phases, and therefore when you mention 11,500 BP hunter-gatherers, were they fewer or had a similar impact than the more recent human groups?” 

Response to Reviewer Point 3: We agree with the reviewer, demography dynamics is important to our discussion. We include demography datum in between lines 370-387. Regarding the impact differences between temporal phases, we expect the marks on the vegetation to be more visible when more recent and / or more persistent. Regarding the persistence of the effects is related to larger population, greater cultural diversity, and more advanced technology. We synthesize the expected effects in a gradient illustrated along the timeline (Fig 1, in line 89).

Reviewer Point 4 “(…) I suggest to consider some aspects to be incorporated in the paper: (…) c) Are there known differences in the cultural choices (environmental-dependent?) of the human groups involved in the research? (probably they are evident from the Fig. 6) Or the differences are also linked to the changes of one culture along the changing millennia?” 

Response to Reviewer Point 4: Yes, there are cultural differences related to the choice of environments. In general, both cultural groups prefer proximity to rivers. But “rivers are especially important to Guarani culture, they are sealers, fishermen” (added in text in lines 293-294). Southern-Jê prefers highland areas, and “Araucaria forests are recognized as an important element in the social organization of the current Southern-Jê (Kaingang and Xokleng/Laklaño)”, i.e, Araucaria forests are key in their social organization (added in lines 288-289). In this way, at the Guarani contact Southern-Jê building monument to defend highland territories (were Araucaria forest occur), marking a cultural transition: “Although there are records of Southern-Jê presence in the highlands since 2000 years BP, it is only around 1000 years BP that these groups started to build their funerary earthworks [61,65]. This new architectural expression may be a way of controlling their territory, because it coincides with the arrival of Guarani groups [65].” (lines 419-422). In the moment of contact with European immigrants, another change in the distribution pattern occurs. Due to the collapse of populations, areas were opened for re-occupation: “Following contact with Europeans, Amerindian populations were reduced due to territorial conflicts, diseases, political, and social disruption. There is evidence of Guarani occupation followed by subsequent Southern-Jê occupation in some locations [25], suggesting that the groups competed for the territories” (lines 424-427). The migration patterns of each group also influence the choice of occupied areas: Guarani along rivers and forming networks, and Southern-Jê in highlands. In manuscript is as follows: “Guarani groups expanded their occupation along the margins of major rivers and along the coastline [30]. The Jê people were distributed over the highlands in their Brazilian territory [25,26], and their pattern of migration created isolated groups, differing from the Guarani [25].” (lines 421-423). 

Reviewer Point 5: “Line 36 = Araucaria can be added to the Keywords.” 

Response to Reviewer Point 5: We agree and include Araucaria in the key words. 

Reviewer Point 6: “Lines 68-69 = how long is this ‘long-term’ history? It emerges from the next paragraph that it is about 11,500 years long. Therefore, a more precise chronology can be added, such as changing “identifying many archaeological sites” to “identifying many Holocene archaeological sites”, or something similar.”

Response to Reviewer Point 6: We added a timeline (Fig 1) and added “Holocene” in text. 

Reviewer Point 7: “Line 103 = “humans are cultural niche constructors” = this statement is absolutely sharable and there is a general consensus on this vision, as well as ‘different cultural groups also leave differing floristic composition legacies’ = please note, the presence of archaeological sites and archaeological/archaeobotanical findings is mandatory to infer the type of land use. Further information can be obtained from the ethnobotanical review, as in the discussion of this paper.” 

Response to Reviewer Point 7: We agree that the description of our hypothesis highlighted aspects that are consensus. We have included an important aspect in the text, about the possibility that there are marks of human actions from the past visible on current forests despite the intense modification in the last century, as in our study area (Sevegnani et al, 2019). In manuscript is as follows: “our hypothesis is that floristic composition differs in Southern Atlantic Forests with a high probability of past human activities, and that different cultural groups also leave differing floristic composition and abundance legacies notable yet, despite the disruption of these cultural groups and of the intense changes caused by recent populations.” (lines 112-116).

The gradient parallel to the timeline in Fig 1 illustrates this.

Reviewer Point 8: “Line 244 = the first part of the discussion is well centred on geographical and geomorphological characteristics that can explain the observed results and trends. However, the timeline variable – chronology of archaeological sites/events is not reported in the following paragraphs.”

Response to Reviewer Point 8: In our analysis, we prioritize the spatial dimension. We ignore the dates because this information is not available for all archaeological sites. Including dates would reduce sampling, making it difficult to calibrate the models. Our results indicate the occupation space accumulated between ~ 3,000-300 years BP. We include this information in the manuscript methods and discussion as bellow: “We do not consider the temporal information of archaeological sites, as this would considerably reduce the sample size.” (lines 156-158) and “Our maps explain the spatial dimension of the occupation of southern-Jê and Guarani, but the temporal dimension mixes the entire period of pre-Colonial occupation.” (lines 412-414).

Reviewer Point 9: “Line 253 = ‘Alternatively, the species could’ = Alternatively, the plant species could.”

Response to Reviewer Point 9: We agree and modify the text.

Reviewer Point 10: “Line 255 = as an example on the need of chronological details, in the sentence ‘Amerindians intentionally cleared the land and used fire to produce crops,’, you should add the culture and/or Holocene period or millennia BP.” 

Response to Reviewer Point 10: We agree and modify the text. We highlighted in text the moment when fire started to be used to open clearings: “In the State of Santa Catarina palynological evidence indicates that the opening of clearings for cultivation started between 3760 and 2430 B.P (Fig 1) [34]” (lines 269-271).

Reviewer Point 11: “Line 264 = “may have involved selective extraction of species for fuel” = is this a speculative argument or can you make an example from a case study?”

Response to Reviewer Point 11: We agree and include study case examples: “Fuel for funeral use must reach very high temperatures, for domestic use it must be easily controllable [63]. The use of fire in rituals can also lead to the selection of woods by symbolic characteristics. Archaeological research has shown that Southern-Jê used preferred species in contexts that indicate the practice of rituals: Araucaria angustifolia, and Jacaranda sp. in a location where Araucária is not available [64].”(lines 278-281).

Reviewer Point 12: “Lines 279-310 = This is an interesting ethnobotanical part of the text, but it is not clear in which order the plants/uses are discussed, and why they are listed.”

Response to Reviewer Point 12: We rearranged the paragraph. Now the text follows a sequence from left to right along the gradient illustrated in Fig 7 (it was Fig 6). That is, starting with species related to the Southern-Jê occupation and ending with species related to the Guarani occupation. The species are listed in Fig 7. We have included captions in the text and in the figure to facilitate the location of each species. For some of the listed species, we discussed ecological and utilitarian features that can help us understand this gradient. We also highlight those species that have an indigenous reference in their names. 

Reviewer Point 13: They “Line 285 = phytolits in pottery could suggest consumption or, preferably, are they evidence of plant processing?” 

Response to Reviewer Point 13: They could be evidence of processing. We modify the text. 

Reviewer Point 14: “Line 385 = practical” 

Response to Reviewer Point 14: We agree and modify the text. 

Reviewer Point 15: “Citations 31. 32, 33, 77 = put ‘Aracauria’ in italics.” 

Response to Reviewer Point 15: We agree and modify the text.

Reviewer Point 16: “Citation 37, 55 = check the format.” 

Response to Reviewer Point 16: We agree and modify the text.

Reviewer Point 17: “Fig. 6 = It is an interesting figure (please, quote it in the text) but I am not sure that it is sufficiently clear the sense and aim of this ‘separation of plant species’ – Cultural? Phytogeographical? Current or past different exploitation? The full name of species should be preferred to the abbreviation of the genus, if possible.”

Response to Reviewer Point 17: We quote Figure 7 (it was 6) in text. The figure shows a list of species with RDA scores above |0.1| and organizes them along a cultural gradient between Southern-Jê and Guarani. Thus, we show species that has its distribution associated with the distribution of archaeological sites Southern-Jê and Guarani. Species can also be related to the environmental characteristics of the archaeological sites’ specific location. They could be yet related with an interaction of environment and occupation. There are many factors that influence the distribution of species and the causes cannot be explained by our analyzes. We highlight this in the manuscript as follow: “An important point is that our analyzes do not isolate the causes of the distribution of plant species, we are considering niche construction as a process in which factors of different natures are inseparable.” (lines 300-301).

---

## [Decision Letter · Decision Letter 1]

8 May 2020

PONE-D-19-33906R1

Pre-colonial Amerindian legacies in forest composition of southern Brazil

PLOS ONE

Dear Dr. Pereira-Cruz,

Thank you for submitting your manuscript to PLOS ONE. After careful consideration, we feel that it only a few changes are necessary that your manuscript can be accepted for publication in PLOS ONE. Therefore, we invite you to submit a revised version of the manuscript that addresses the points raised during the review process.

We would appreciate receiving your revised manuscript by Jun 22 2020 11:59PM. To enhance the reproducibility of your results, we recommend that if applicable you deposit your laboratory protocols in protocols.io, where a protocol can be assigned its own identifier (DOI) such that it can be cited independently in the future. For instructions see: http://journals.plos.org/plosone/s/submission-guidelines#loc-laboratory-protocols

We look forward to receiving your revised manuscript.

Kind regards,

André Jasper, Ph.D.

Academic Editor

PLOS ONE

Additional Editor Comments (if provided):

Dear Dr. Pereira-Cruz and colleagues,

After carefully revising your resubmission, we are quite reaching a final decision that is now conditioned to a few details to be changed in your manuscript.

Both reviewers and I agreed that your manuscript is an interesting contribution for PlosOne and, for final acceptance, we kindly ask for the following (as pointed out by Dr. Mercuri):

Line 80 = instead of “Zea Mays, Manihot Esculenta, and Dioscorea sp,” = “Zea mays, Manihot esculenta, and Dioscorea sp.”

Line 123 = delete comma after ‘ (SC) ‘

Lines 157-158 = “We do not consider the temporal information of archaeological sites, as this would considerably reduce the sample size.” This sentence was added now to reply to comments by the reviewer, however it may introduce other criticisms: the chronology is a must in archaeology. Probably, it may be better changing the sentence by reporting only the information on the range of chronology of the sites studied, that is: “ the archaeological sites have been occupied during the last XX millennia”

Line 269 = “unintentionally favored certain plants and/or crops by…” = ‘certain’ is vague and, if correct, ‘synanthropic’ (or the name of the species) should be preferred

Line 359 = check “and we need”

Fig. 4 = check the green colors as it seems that the color of Brazil and the of S. America are inverted, especially in the part ‘ c ‘ of the figure.

Looking forward for you return,

André

Reviewers' comments:

Reviewer's Responses to Questions

**Comments to the Author**

1. If the authors have adequately addressed your comments raised in a previous round of review and you feel that this manuscript is now acceptable for publication, you may indicate that here to bypass the “Comments to the Author” section, enter your conflict of interest statement in the “Confidential to Editor” section, and submit your "Accept" recommendation.

Reviewer #1: All comments have been addressed

2. Is the manuscript technically sound, and do the data support the conclusions?

Reviewer #1: Yes

3. Has the statistical analysis been performed appropriately and rigorously? 

Reviewer #1: I Don't Know

4. Have the authors made all data underlying the findings in their manuscript fully available?

Reviewer #1: Yes

5. Is the manuscript presented in an intelligible fashion and written in standard English?

Reviewer #1: Yes

6. Review Comments to the Author

Reviewer #1: The clarity of the paper was notably improved, Fig. 1 is very useful and well done; also the Fig. 7 is now more useful. This research is a very good contribution to the field and I support its publication on PLOSone.

Few remarks are reported below.

Line 80 = “Zea Mays, Manihot Esculenta, and Dioscorea sp,” = “Zea mays, Manihot esculenta, and Dioscorea sp.”

Line 123 = delete comma after ‘ (SC) ‘

Lines 157-158 = “We do not consider the temporal information of archaeological sites, as this would considerably reduce the sample size.” This sentence was added now to reply to comments by the reviewer, however it may introduce other criticisms: the chronology is a must in archaeology. Probably, it may be better changing the sentence by reporting only the information on the range of chronology of the sites studied, that is: “ the archaeological sites have been occupied during the last XX millennia”

Line 269 = “unintentionally favored certain plants and/or crops by…” = ‘certain’ is vague and, if correct, ‘synanthropic’ (or the name of the species) should be preferred

Line 359 = check “and we need”

Fig. 4 = check the green colours a sit seems that the colour of Brazil and ithe S. America are inverted, especially in the part ‘ c ‘ of the figure.

7. PLOS authors have the option to publish the peer review history of their article (what does this mean?). If published, this will include your full peer review and any attached files.

Reviewer #1: Yes: Anna Maria Mercuri

---

## [Author Response · Author response to Decision Letter 1]

22 Jun 2020

Response to reviewer:

We thank the Dr. Mercuri for the assessment and comments of our manuscript. We have modified the text and figures according to the referees’ critiques. Below are our comments on each issue raised by Dr. Mercuri.

Reviewer Point 1: “Line 80 = “Zea Mays, Manihot Esculenta, and Dioscorea sp,” = “Zea mays, Manihot esculenta, and Dioscorea sp.””

Response to Reviewer Point 1: We fix it.

Reviewer Point 2: “Line 123 = delete comma after ‘(SC)‘”

Response to Reviewer Point 2: We delete the comma.

Reviewer Point 3: “Lines 157-158 = “We do not consider the temporal information of archaeological sites, as this would considerably reduce the sample size.” This sentence was added now to reply to comments by the reviewer, however it may introduce other criticisms: the chronology is a must in archaeology. Probably, it may be better changing the sentence by reporting only the information on the range of chronology of the sites studied, that is: “the archaeological sites have been occupied during the last XX millennia””.

Response to Reviewer Point 3: We agree about the high importance of chronology in archaeology and modify the text as suggested. We removed the sentence that might sound while minimizing the importance of the chronology: “We do not consider the temporal information of archaeological sites, as this would considerably reduce the sample size.”. And we include the sentence “The selected archaeological sites have been occupied over the past 3,000 years” (line 159-160).

Reviewer Point 4: “Line 269 = “unintentionally favored certain plants and/or crops by…” = ‘certain’ is vague and, if correct, ‘synanthropic’ (or the name of the species) should be preferred”.

Response to Reviewer Point 4: We agree and remove the word “certain”. We include an example: “… and unintentionally favored plants by interrupting seeds dormancy, like Mimosa scabrella [59–61], by improvement in light availability to pioneer species, like Solanum mauritianum [60].”.

Reviewer Point 5: “Line 359 = check “and we need””. 

Response to Reviewer Point 5: We fix it.

Reviewer Point 6: “Fig. 4 = check the green colours a sit seems that the colour of Brazil and it he S. America are inverted, especially in the part ‘ c ‘ of the figure.”

Response to Reviewer Point 6: We fix it.

---

## [Editor Report · Decision Letter 2]

24 Jun 2020

Pre-colonial Amerindian legacies in forest composition of southern Brazil

PONE-D-19-33906R2

Dear Dr. Cruz and Colleagues,

We’re pleased to inform you that your manuscript has been judged scientifically suitable for publication and will be formally accepted for publication once it meets all outstanding technical requirements.

Kind regards,

André Jasper, Ph.D.

Academic Editor

PLOS ONE

---

## [Editor Report · Acceptance letter]

8 Jul 2020

PONE-D-19-33906R2 

Pre-colonial Amerindian legacies in forest composition of southern Brazil 

Dear Dr. Pereira Cruz:

I'm pleased to inform you that your manuscript has been deemed suitable for publication in PLOS ONE. Congratulations! Your manuscript is now with our production department. 

Kind regards, 

on behalf of

Prof. Dr. André Jasper 

Academic Editor

PLOS ONE